# Quality and Authenticity Control of Functional Red Yeast Rice—A Review

**DOI:** 10.3390/molecules24101944

**Published:** 2019-05-20

**Authors:** Jiawen Song, Jia Luo, Zubing Ma, Qiang Sun, Chunjie Wu, Xiaofang Li

**Affiliations:** School of Pharmacy, Chengdu University of Traditional Chinese Medicine, Chengdu 611137, China; Songjiawen@stu.cdutcm.edu.cn (J.S.); ljlj6610@sina.com (J.L.); 18280322839@163.com (Z.M.); sunqiang@stu.cdutcm.edu.cn (Q.S.)

**Keywords:** functional RYR, monacolin K, detection methods, quality standards, authentication methods

## Abstract

Red yeast rice (RYR) is made by fermenting the rice with *Monascus*. It is commonly used in food colorants, dyeing, and wine making in China and its neighboring countries. Nowadays RYR has two forms on the market: common RYR is used for food products, the other form is functional RYR for medicine. However, some researchers reported that commercial lovastatin (structure is consistent with monacolin K) is illegally added to common RYR to meet drug quality standards, so as to imitate functional RYR and sell the imitation at a higher price. Based on current detection methods, it is impossible to accurately distinguish whether functional RYR is adulterated. Therefore, it is especially important to find a way to authenticate functional RYR. In the current review, the advances in history, applications, components (especially monacolins, monacolins detection methods), quality standards, authentication methods and perspectives for the future study of RYR are systematically reviewed.

## 1. Introduction

### 1.1. The History and Development of RYR

RYR, which is solid-fermented by artificial inoculation of *Monascus purpureus* on steamed rice [1], has been widely used for thousands of years by China and its neighbors. It has various names in different cultures [2]. For example, it is called Chiqu, Hongqu, Red rice, Fuqu, etc., in China; Koji, AngKhak, Beni-Koji, and Red-Koji in Japan; Rotschimmelreis in Europe; and Red Mold or Red mold rice in the USA. In ancient times, Fujian, Zhejiang, and Taiwan were considered to be the places of origin of RYR, especially Gutian (a town in Fujian province) which has been confirmed as the genuine origin of RYR because of the high quality and production there [3,4]. The earliest record of RYR has it being used as a food flavoring, colorant, and brewing agent, such as in fermented bean curd, pork stew, roast meat products, and rice wine [5]. Subsequently, the ancients discovered that RYR has therapeutic effects. The therapeutic effects have been recorded in the *Compendium of Materia Medica*, an ancient Chinese pharmacopoeia written by Li Shizhen during the Ming dynasty [6], which narrated in detail how RYR has “the effect of promoting the circulation of blood and releasing stasis, invigorating the spleen and eliminating digestion.”

Since 1979, monacolin K, which can lower lipids in blood, has been extracted from *Monascus* species [7]. Nowadays, RYR comes in two forms on the market [8]: common RYR, which is used as a source of natural pigments for coloring in the food and textile industries. The fermentation conditions of common RYR are relatively quick and simple. According the “Food additives Red yeast rice GB1886.19-2015,” the quality of common RYR is evaluated by color [9]. The other type is called functional RYR, which has a definite lipid-lowering effect. The fermentation conditions of functional RYR need to be optimized scientifically, including through screening of *Monascus* strains and determination of optimal fermentation conditions (temperature, moisture content, time, etc.). The quality of the functional RYR is evaluated by the contents of monacolin K, which is one of the secondary metabolites of *Monascus* [10,11]. Because of the well-known lipid-lowering efficiency of functional RYR, there are many Chinese patent medicines containing functional RYR for reducing lipids on sale in the market, such as Zhibituo, Xuezhikang and Lezhiping [12,13,14].

### 1.2. Monacolins in RYR

RYR contains several types of secondary metabolites that have biological activity, including *Monascus* pigments, monacolins, gamma-aminobutyric acid, dimerumic acid, enzymes, polysaccharides, ergosterol, polyketides, unsaturated fatty acids, phytosterols, isoflavones, alkaloid, and trace elements [15,16,17]. Secondary metabolites in RYR have been shown to have biological activities through modern pharmacological experiments. For instance, *Monascus* pigments (MPs) are usually used as colorants in the food and textile industries, but research has also demonstrated that they can have effects such as anti-obesity, lipid-lowering and attenuating nonalcoholic fatty liver disease in mice [18,19,20]. On the other hand, in RYR, there is a compound named citrinin that is a highly toxic, mutagenic, and carcinogenic metabolite [21]. It is worth noting that monacolin K has been researched in depth [22].

#### 1.2.1. Physical and Chemical Properties of Monacolin K

Monacolin K is the first reported lipid-lowering component to be isolated from *Monascus purpureus* [23]. Monacolin K, which appears as colorless crystals, is soluble in methanol, ethanol, acetone, chloroform, and benzene but not soluble in n-hexane or petroleum ether. The boiling point is 157–159 °C and the ${\left[\alpha \right]}_{D}^{25}$ value is +307.6 °C (in methanol). The molecular formula is C_24_H_36_O_5_ (Mw 404) and the ratio of C:H:O is 71.31:8.91:19.78, obtained by elemental analysis and high-resolution mass spectroscopy. The Rf value in TLC is 0.47 in dichioromethane: acetone (4: 1, ***v***/***v***). The UV spectrum (methanol) showed maxima at 229, 237, and 246 nm. The IR spectrum (KBr) showed absorption bands at 3550, 2970, 1696, and 1220 cm^−1^. The ^13^C-NMR spectrum (CD_3_OD) indicated the presence of two ester carbonyl carbons, four methyl carbons and methylene and methine carbons. The mass spectrum peaks (*m*/*e*) are at 404 (M^+^), 302 (M − 102), 284 (M − 120), and 224 (M − 180), and prominent peaks in the mass spectrum of monacolin K were observed at 198 (M − 206), 172 (M − 232), 159 (M − 245), and 157 (M − 247) [7]. There are two chemical structures in functional RYR, shown in Figure 1, the acid and lactone forms of monacolin K.

#### 1.2.2. The Mechanism of Lipid Lowering of Monacolin K

The mechanism of lipid lowering of monacolin K is shown in Figure 2 [24]: competitively inhibiting endogenous cholesterol synthesis speed limit enzyme (3-hydroxy-3-methylglutaryl-coenzyme A (HMG-CoA) reductase, a key enzyme in cholesterol biosynthesis in animals and humans), blocking the pathways of hydroxyl valeric acid metabolic in cells, reducing cholesterol synthesis in the cell, stimulating the quantity and activity of LDL receptors on the cell membrane surface, and removing the serum cholesterol. RYR can also inhibit the absorption of cholesterol, and the active ingredient may be sterols [24]. In the procedure, the mean serum total cholesterol (TC) and triglyceride (TG) levels decline and the low-density lipoprotein (LDL, the bad lipoprotein) level decreases significantly, but the high-density lipoprotein (HDL, the good lipoproteins) level does not change much [25]. The acid form is the efficient form because its structure is similar to that of HMG-CoA, shown in Figure 3, and can directly inhibit the synthesis of cholesterol [26]. Therefore, the lactone form needs to be hydrolyzed into an acid structure by the action of the hydroxyl esterase in the body, and then to be functionalized.

#### 1.2.3. Monacolin Analogs

In addition, there are reports showing that there are other monacolins in RYR [27]. Evaluating the activity of newly discovered compounds in vitro, new monacolins extracted from RYR had a similar lipid-lowering effect to monacolin K [28]. In the 1980s, Endo isolated monacolin J, L, X, M, and dihydromonacolin L from *Monascus purpureus*, and proved that all of them also could inhibit HMG CoA reductase and reduce the serum LDL and TC levels [29,30,31]. In 2004, Li et al. [32] identified 14 monacolin compounds such as monacolin K, J, L, M, X, and their hydroxy acid form, as well as dehydromonacolin K, dihydromonacolin L, compactin, 3α-hydroxy-3,5-dihydromonacolin L, etc. in RYR. In 2006, Dhale et al. [33] separated dihydromonacolin MV in RYR. In 2011 and 2013, Liu et al. [34,35] isolated monacophenyl, monacolin O and P from RYR. In 2012, Zhu et al. [36] extracted two new dehydromonacolins (dehydromonacolin N, J) and detected 10 other monacolins (α,β-dehydrodihydromonacolin K, L, and the ethyl ester of monacolin K, etc.) from RYR. In 2016 Zhang et al. [15] obtained new five monacolins, including monacolin R, α,β-dehydro-monacolin Q, monacolin Q, α,β-dehydro-monacolin S, and monacolin S. In 2017, Zhang et al. [37] found four new monacolins, including monacolin T, monacolin U, 6*a*-*O*-methyl-4,6-dihydromonacolin L, and 6*a*-*O*-ethyl-4,6-dihydromonacolin L. The chemical structures of the monacolins investigated in RYR are shown in Table 1. What is even more remarkable is that Li et al. [38] isolated more than 80 kinds of monacolins from the RYR product Xuezhikang and established a database of more than 720 potential monacolins, providing a potential effective method for the discovery of new analogous series, and offering a material basis for the further exploitation of RYR products.

### 1.3. The Biosynthetic Pathway and Sources of Monacolin K

There are few studies about the biosynthetic pathway of monacolin K generated by *Monascus purpureus*, it is just speculated that this pathway may be similar to the synthetic pathway of monacolin K in *Aspergillus terreus* [39]. *Aspergillus terreus* has been frequently utilized to figure out the synthesis of monacolin K [40]. The synthetic pathway of monacolin K in *Aspergillus terreus*, shown in Figure 4 is as follows: (1) the lovastatin nonaketide synthase (LNKS) catalyzes the synthesis of one molecule of malonyl CoA in condensation reaction with nine molecules of acetyl CoA, generating dihydromonacolin L (a nonaketide compound) that is a main structure of monacolin K, then it formed monacolin L by oxidation, dehydration, and other steps, and finally monacolin J is generated through single oxygenase catalytic hydroxylation reaction; (2) the lovastatin diketide synthase (LDKS) catalyzes the condensation of one molecule of malonyl CoA with two molecules of acetyl CoA to form methylbutyryl CoA; (3) under the action of the transesterase, methylbutyryl CoA is linked to monacolin J with an ester ether bond to complete the synthesis of monacolin K.

Meanwhile, commercial lovastatin appeared as a lipid-lowering drug in the pharmaceutical market with the permission of the FDA in 1986. The sources of commercial lovastatin mainly contain two synthetic approaches. The first is that utilize submerged fermentation (SmF) of *Monascus purpureus* and *Aspergillus terreus*, adding carbon and nitrogen source (such as glucose, oat meal, and corn steep liquor) to medium with *Monascus purpureus* and *Aspergillus terreus*, after appropriate fermentation, then obtaining lovastatin via separation and purification; the other is chemical synthesis, replacing the decaline ring of the fungal compounds with an aromatic ring, like fluvastatin and atorovastatin [41,42]. Early studies have shown that the structure of monacolin K is chemically identical to that of commercial lovastatin [32].

In common RYR, monacolin K cannot be produced or is very tiny in a short and simple process of fermentation. Recently researchers reported that commercial lovastatin is added to common RYR to imitate functional RYR and therefore obtain more profits [43,44]. This phenomenon has inhibited the development of functional RYR, and more important, long-term use of lovastatin probably can cause burnout, gastrointestinal reactions, myalgia, etc. About 10–15% patients with dyslipidemia who take statins experience skeletal muscle problems [11].

## 2. The Detection Methods of Monacolin K of RYR

Due to the lipid-lowering activity, monacolin K is used as a quality control indicator of RYR. The detection methods for monacolin K in RYR are shown in Table 2, mainly including HPLC, HPLC-MS/MS, GC-MS, and so on, and the analysis instrument, measurements conditions, and detection indicators are described in detail.

In HPLC analysis, the separation and detection of species of several elements in a single analytical run can be accomplished. In order to make HPLC analysis simpler and more efficient, it is often coupled with UV and DAD detectors. For research into RYR products, HPLC is a common detection method and most laboratories and manufacturers can satisfy this condition [75]. As can be seen from the data in Table 2, the detection method of monacolins in RYR is mostly HPLC and the detection index of RYR is mostly monacolin K lactone. For monacolin K acid and other monacolins, acetonitrile: 0.1% trifluoroacetic acid, acetonitrile: 50 mM potassium hydrogen phosphate and methanol: 0.1% aqueous phosphoric acid were used as the mobile phase. Tsukahara [47] used methanol: 0.1% aqueous phosphoric acid (72:28, *v*/*v*) as the mobile phase, isocratic elution for 25 min, and linear elution with 100% methanol until 30 min, respectively. The contents of monacolin K (lactone) and citrinin were detected; *Monascus sp*. with a deep red color, high monacolin K production, and no citrinin was screened. Although there are many HPLC methods, the measurement conditions such as a mobile phase, separation column, and detector are different for each research purpose, resulting in different detectable components. Zhang et al. [48,49] developed a new SSF process using agar as a carrier and then optimized the grain for the highest yield of monacolin K by SSF with determination of monacolin K by HPLC, and the effects of fermentation substrate, fermentation temperature, and time on the yield of monacolin K were investigated. Liu et al. [50] used HPLC-UV to optimize the extraction conditions of monacolin K in functional RYR.

MISPE is a solid-phase extraction technique based on molecular imprinting which is often combined with HPLC and HPLC-MS/MS. It can significantly improve the extraction rate and purity of monacolin K in RYR, and reduce the impurity of the components to be tested. The matrix effects on the analysis results can be removed, and the subsequent analysis of monacolin K was more efficient and faster. Eren et al. [58] developed a sensitive molecular imprinted quartz crystal microbalance (QCM) sensor for the selective determination of monacolin K lactone in RYR.

GC-MS/MS and HPLC-MS/MS are joint detection techniques. In the actual analysis, HPLC-MS/MS is more commonly used. The tandem mass spectrometry technology has higher resolution and a faster analysis speed. It offers good detection of monacolin analogs in RYR. Donna et al. [68] uses the multiple reaction monitoring (MRM) mode and HPLC-MS/MS as the analytical method: simvastatin and pravastatin were used as indicators to determine the acid and lactone monacolin K content in RYR. Different detectors have different detection ranges. Svoboda et al. [69] connected solid-phase extraction of molecularly imprinted polymers, developing a UHPLC-MS method without a matrix effect. Mornar et al. [70] connected diode arrays, fluorescence, and mass spectrometry detectors to simultaneously measure monacolin K, L, M, monacolin K hydroxy acid, dehydromonacolin K, and citrinin in RYR products.

In addition, Nigovic et al. [59] developed a simple and fast voltammetric method to detect the content of lactone monacolin K in RYR, and used micellar electrokinetic capillary chromatography to detect the two configurations of monacolin K in red yeast preparations [76].

## 3. The Standards of Quality Control for RYR

Due to the lack of scientific data, RYR has not been officially registered by the FDA [77]. Therefore, RYR is often used as a food or as a dietary supplement abroad. Taiwan promulgated the Hongqu Health Food Standard in 2007 (Taiwan Guardian Food No. 0960406448). Regarding the red blood safety index, the national standard for the detection of citrinin is “Determination of citrinin in RYR products (GB T5009. 222-2008).” The European Food Safety Authority is currently reassessing the safety of a 10-mg dose of monacolin K as a food supplement [78].

In addition, according to a literature search, only the *British Pharmacopoeia* provides the functional RYR standard abroad. Most of the standards are the laws of various provinces and cities in China. In China, monacolin K is the lipid-lowering component in functional RYR, so the pharmacopoeia, ministerial standards, and regional standards evaluate the quality of functional RYR based on the content of monacolin K lactone. The current standards are listed in Table 3.

According to Table 3, most standards have no clear requirement for inspection of the content of monacolin K of RYR; some of the above criteria clarify the required content of monacolin K lactone in RYR (generally not less than 0.40%), while just a few standards mentioned the content requirements of monacolin K lactone and acid. In Table 3, only five standards have been specified for the determination of monacolin K in functional RYR. In the *Chinese Pharmacopoeia* (2015), monacolin K lactone is the only index of quality control of functional RYR and the limit of minimum content of monacolin K lactone is 0.22%. “The standard of Chinese herbal medicine of Yunnan Province (2005)” and “Standard for Chinese Medicine Yinpian Processing of Sichuan Province (2015)” also required that monacolin K lactone should serve as the quality control of functional RYR and the lowest content is 0.40%. However, there are two standards that indicate requirements for the acid structure. In the industrial standard “Functional red yeast rice QB/T 2847-2007,” it is regulated that the sum of monacolin K lactone and acid in functional RYR should be not less than 0.40%. The “Standard for Chinese Medicine Yinpian Processing of Zhejiang Province (2015)” required that in functional RYR, the sum of monacolin K lactone and acid must be more than 0.30% and the peak area of acid monacolin K must not be less than 5% of the lactone monacolin K peak area. In addition to the above criteria, the other standards listed in Table 3 do not mention the quality control components and content limits of functional RYR.

*Monascus* can produce harmful secondary metabolites during the fermentation process, such as citrinin, which is a highly toxic, mutagenic, and carcinogenic metabolite and has been implicated as a causative agent in human hepatic and extrahepatic carcinogenesis [21]. It causes serious health problems such as liver and kidney disease, nervous system damage, and carcinogenicity. In addition, aflatoxin may be generated, caused by bacterial contamination in the fermentation process, for a variety of reasons. Aflatoxins are potent carcinogens that affect all organ systems, particularly the liver and kidneys, and are also genotoxic and may cause birth defects in children. They can cause immunosuppression and may reduce resistance to infectious agents such as HIV and tuberculosis. The ingestion of citrinin and aflatoxins is harmful to humans and animals. Therefore, it is necessary to carry out a safety check on the RYR. There are three standards that require security checks. For the determination of citrinin, “Standard for Chinese Medicine Yinpian Processing of Sichuan Province (2015)” and “Functional red yeast rice QB/T 2847-2007” use the HPLC method; detection is by a fluorescence detector with excitation wavelength λ_ex_ = 331 nm, emission wavelength λ_em_ = 500 nm. The two standards decree that the content of citrinin should not exceed 50 μg/kg. The ‘’Food additives Red yeast rice GB1886.19-2015′’ and “Standard for Chinese Medicine Yinpian Processing of Sichuan Province (2015)” utilize two methods to detect the content of aflatoxins. One is HPLC with a fluorescence detector (λ_ex_ = 360 or 365 nm, λ_em_ = 450 nm). The other is HPLC-MS with ESI, in positive ion mode, with triple quadrupole tandem mass spectrometry as the detector. The two standards require that the content of aflatoxin B1 not exceed 5 μg/kg.

## 4. The Authentication Methods of Functional RYR

Functional RYR has a large market as a lipid-lowering drug, with the indicator ingredients being monacolin K. Because the structure of monacolin K in RYR is naturally the same as that of commercial lovastatin [12], it is not possible to distinguish commercial lovastatin from monacolin K with the currently used detection methods, such as HPLC, HPLC-MS/MS, and GC-MS. Some researchers [100,101] have found that commercial lovastatin is added to common RYR to impersonate functional RYR and gain illegal profits in the pharmaceutical market. Therefore, it is necessary to find a reasonable and effective method for authenticating functional RYR as quickly as possible. There are a few reports of a method for authenticating functional RYR, shown in Table 4.

Zhu et al. [8] extracted decalins from different types of RYR, including four batches of common RYR, two batches of functional RYR, and three functional RYR formulas (Xuehzikang capsule, Hongqu Xiaohisu, and Zhibituo tablets), and simultaneously determined their levels by UHPLC–QQQ-MS and UHPLC-Q-TOF-MS. The chromatography was performed on an ACQUITY UHPLC BEH C18 column (100 mm × 2.1 mm, 1.7 μm), with formic acid and acetonitrile as the mobile phase with gradient elution, a flow rate of 0.35 mL/min, and a column temperature of 40 °C. The results of UHPLC–QQQ-MS analyses showed that the individual or total decalin contents in the raw material of functional RYR are much higher than its preparations. However, no decalin was detected in the common red yeast rice. The data demonstrated that decalins exist in functional RYR but not in common RYR. Moreover, based on the quantitative results of decalins in RYR, heptaketide was proposed to be another marker for the quality control of functional RYR. Their study demonstrated that the UHPLC-Q-TOF-MS and UHPLC–QQQ-MS methods described in their paper are powerful and reliable tools for the quality control of RYR.

Nannoni et al. [100] found that monacolin K and monacolin K acid form (MK and MKA are considered to be the active monacolins) have a proportional relationship in all secondary monacolins, which can be used to determine whether commercial lovastatin was added to RYR. The instrument conditions: Spherisord ODS-2 column (250 × 0.4 mm, 0.5 μm + precolumn Zorbax Reliance Cartridge), DAD detector at 237 nm, phosphoric acid and acetonitrile as the mobile phase with gradient elution; the flow rate is 1 mL/min. When analyzing the data, it was found that there was a scaling relationship between MK and MKA and the sum of the total secondary monacolins. Two discriminant equations had been proposed as follows. If the proportional relationship of RYR products did not meet the two discriminants as mentioned, the product is determined to be adulterated; if only Equation (1) did not accord, it proves that it is likely adulterated; if only the Equation (2) did not accord, it proves that it is possibly adulterated. MK and MKA tend to transform each other, and the content of MK is dominant, but does not exceed 70% of the total of the two monacolins. Therefore, if the MK content is higher in the HPLC analysis, it is proven that the product is likely adulterated. In 32 samples’ data, 16 did not meet the formula, which indicated that the 16 samples were likely to be adulterated.
(1)$$\frac{\mathrm{other}\;\mathrm{secondary}\;\mathrm{monacolins}}{\mathrm{MK}+\mathrm{MKA}}>0.02$$
(2)$$\frac{\mathrm{MKA}}{\mathrm{MK}+\mathrm{MKA}}>0.30$$

Perini et al. [101] utilized the analysis of δ^2^H and δ^13^C with Isotope Ratio Mass Spectrometry to distinguish monacolin K in functional RYR and commercial lovastatin. The fermentation matrix of SSF and SmF of RYR is different. Commercial lovastatin is produced by SmF using a *Aspergillus terreus* mutant, which decreases the production costs. In general, sufficient nutrients such as glucose, oat flour, corn steep liquor, polyethylene glycol, and trace elements are administered on the culture medium, and different substrates including glucose, lactose, corn, etc. are used as the media of SmF. SSF mainly uses cereals like millet as a fermentation substrate for the production of MK in functional RYR. In ^1^H-NMR analysis, the δ^2^H values of 15 commercial lovastatin and three drugs with lovastatin ranged from −207.7‰ to −265.5‰; 18 monacolin K from functional RYR ranged from −245.6‰ to −269.2‰. In ^13^C-NMR analysis, the δ^13^C values of 15 commercial lovastatin and three drugs with lovastatin ranged from −15.3‰ to −21.4‰; 18 monacolin K from functional RYR ranged from −30.7‰ to −28.2‰. The results showed that ^1^H-NMR cannot identify the authenticity of functional RYR, but ^13^C-NMR can. The δ^2^H values in samples are mainly derived from water, and, due to different water sources in different regions, the δ^2^H values are different. The δ^13^C values of MK extracted from functional RYR and commercial lovastatin have a completely different δ^13^C range because of the different matrices. These phenomena are consistent with the biological sources in the matrix; the rice has a C3 photosynthesis cycle with a δ^13^C value between −27‰ and −25‰. The plant photosynthesis cycle of C4 is in the range of −10‰ to −16‰, which is consistent with the determination of lovastatin [102]. Moreover, MK would not undergo isotope fractionation during extraction. All of the above demonstrated that the analysis of δ^13^C with isotope ratio mass spectrometry can authenticate functional RYR.

## 5. Future Perspectives and Conclusions

Nowadays, increasing evidence has demonstrated that monomers/extracts derived from Traditional Chinese Medicines (TCM) are valuable resources for finding novel and effective candidate drugs with low toxicities [103,104]. RYR, a known TCM fermented by *Monascus*, could be used to stimulate blood circulation, the spleen, and digestion. Since monacolin K was first isolated from *Monascus* and determined by Endo to have a lipid-lowering effect, *Monascus* fermentation products have attracted much attention, and RYR is one of the most eye-catching products, divided into common RYR for food and functional RYR for medicine in the market.

Monacolin K (lactone and acid) is the main lipid-lowering component of functional RYR, and the acid form is the efficient form in the body. At present, many studies have shown that there are many other monacolins, including dihydromonacolin MV, monacophenyl, monacolin O and P, etc. in RYR, and some of them have similar effects to monacolin K.

Since the structure and biosynthetic pathway of monacolin K are similar to that of commercial lovastatin, it is hard to differentiate the two using common detection methods, such as HPLC, HPLC-MS/MS, and GC-MS. Therefore, some researchers have reported that commercial lovastatin is added to common RYR to imitate functional RYR, satisfying the quality standard requirements of functional RYR and thus achieving illegal profitability. This has had a negative impact on the RYR market. In response to this phenomenon, there are a few studies that have shown that some suitable methods can be used to authenticate functional RYR. Although there are currently three methods for identifying functional RYR, in order to better identify functional RYR, researchers should conduct more in-depth research and establish a practical, accurate, fast, and stable authentication method.

The composition of TCM is complex, and how to control the quality of TCM is becoming a hot topic. The quality evaluation of TCM is the basis for ensuring the effectiveness and safety of TCM. Due to a variety of factors, the functional RYR quality would be affected, so a dependent quality control system needs to be established to ensure the highest degree of safety and effectiveness for functional RYR products [105]. For solving this problem, more complete quality control standards of TCM should be established.

### 5.1. The Spectrum-Effect of TCM

The spectrum-effect of TCM, due to its comprehensive and quantifiable advantages, has been widely used in the quality control of single-flavor and Chinese medicine formulas in recent years [106]. The spectrum-effect study of TCM aims to establish a link between the fingerprint and the actual efficacy; according to the results of pharmacological experiments, the chromatographic peaks related to the efficacy of the drug are found and structurally confirmed, so as to clarify the basis of the pharmacodynamic substance. This achieves the purpose of controlling the quality of TCM [107]. Shi et al. [108], using UPLC-MS^2^ analysis, identified the psoralen ethanol extract and determined the tyrosinase activity by the oxidative rate method of levodopa, evaluating its inhibition and the activation of tyrosinase. Based on the anti-inflammatory and analgesic activity of *Alpinia zerumbet*, Xiao et al. [109] established the fingerprint of *Alpinia zerumbet* volatile oil and simultaneously evaluated the anti-inflammatory and analgesic effects of *Alpinia zerumbet* volatile oil by pharmacological experiments. After spectral-effect correlation assessment, the main bioactive components of analgesic and anti-inflammatory activities may be α-furan, β-pinene, camphor, and α-cadino. The lipid-lowering components of functional RYR include monacolin K, J, L, M, X, and their acid forms, etc., but the current standards either have no indicator components or only determined monacolin K; therefore, in order to truly reflect the quality of functional RYR, according to the spectral pharmacodynamics, drug supervision and administration departments should establish characteristic fingerprints of functional RYR to better evaluate its quality.

### 5.2. The TCM Quality Traceability System Based On the Quality Marker

The TCM quality traceability system of functional RYR should be established based on the “quality marker” (Q-marker) [110]. The TCM quality traceability system uses Internet of Things and block chain technology, cloud computing, big data, and other information technologies to process the key information of TCM from planting, processing, production, and circulation to use, and then to ensure that sources of TCM can be known and traced, its quality can be checked, and responsibility can be investigated [111,112]. Q-marker is a complex system for quality assessment and production process control of TCM products with traceability. Q-marker reflects the compatibility of TCM and modern pharmacological study; the drug effect (such as effectiveness and safety) should be demonstrated to be associated with the identified quality marker [113]. Sheng [114] designed an electronic products code (EPC) of quality traceability for Bobaishao Decoction pieces based on a radio frequency identification (RFID) tag, and applied it to the quality traceability Internet of Things system of Bobaishao Decoction pieces. Zhou et al. [115] determined that the quality markers for *Bulbus fritillariae* were 28 compounds such as solanidine, zhebeiresinol, puqiedine, etc. by using UPLC-Q/TOF-MS combined with principal component analysis (PCA), an artificial neural network (ANN), and biological evaluation. The TCM quality traceability system based on the Q-marker has received more and more attention in the quality control of TCM. However, the formal establishment of this system requires a lot of related research, such as Q-marker selection methods (biopotency, biological fingerprinting, pharmacokinetics, etc.), design methods of traceability system, and processing of information technology such as block chain, cloud computing, big data, etc.

### 5.3. Improvement of Relevant Policies and Regulations of Functional RYR

With the deepening of the study of functional RYR, it is found that monacolins play a synergistic role in the lipid-lowering effect of functional RYR [24]. However, the current standards use monacolin K as the sole indicator of quality control of functional RYR. Therefore, the quality of the RYR products cannot be truly reflected by the current standards.

Based on existing detection methods, in order to reflect the quality of the functional RYR, the content and ratio of acid and lactone monacolin K should be determined synchronously. Then we can gradually establish better quality control methods, such as characteristic fingerprints based on spectrum-effect research. In addition, the quality control of drugs is inseparable from the help of government departments. Relevant departments should strengthen the supervision of the RYR market, strictly control the application approval of RYR producers, stipulate the fermentation process of different types of RYR, including the production conditions such as strains, fermented grains, fermentation time, etc., and check the functional RYR on the market for quality. For comprehensively evaluating the quality of functional RYR, it is necessary to establish better quality standards with a strong policy-oriented nature; at the same time, policy regulations are required to crack down on illegal substitutions.

## Figures and Tables

**Figure 1 molecules-24-01944-f001:**
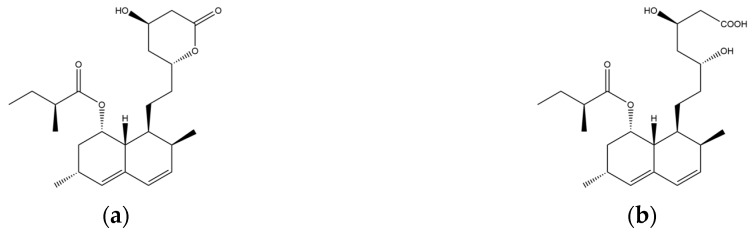
The two structures of monacolin K. (**a**) the lactone form; (**b**) the acid form.

**Figure 2 molecules-24-01944-f002:**
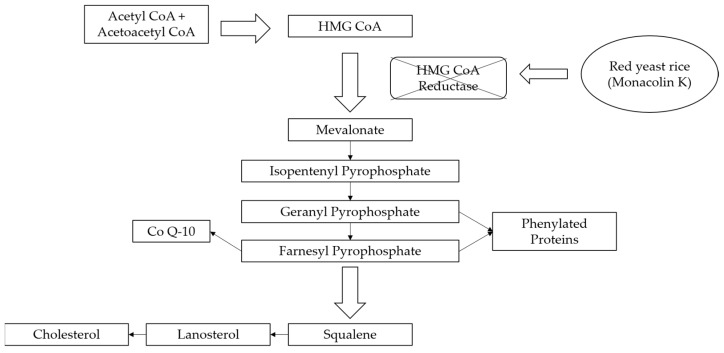
RYR effects on cholesterol biosynthesis.

**Figure 3 molecules-24-01944-f003:**
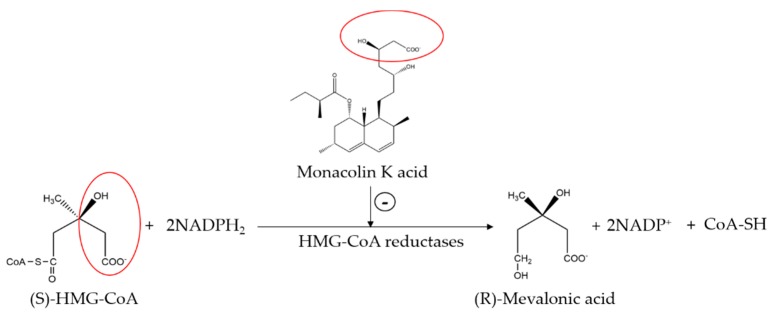
The HMG-CoA reductase reaction and the competitive inhibition of monacolin K acid.

**Figure 4 molecules-24-01944-f004:**
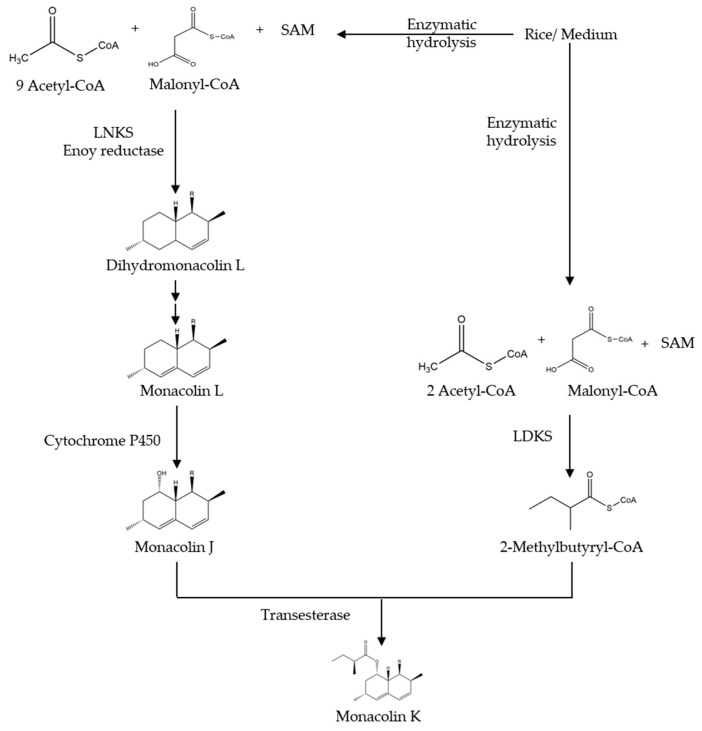
The biosynthetic pathway of Monacolin K. Abbreviations: SAM, *S*-Adenosyl methionine; R, 2-[(2*R*,4*R*)-4-hydroxy-6-oxo-2-tetrahydropyranyl]-ethyl.

**Table 1 molecules-24-01944-t001:** Chemical structures of the investigated monacolins in RYR.

Name	Type	Structure	Formula	MW	Activity	Ref.
Monacolin	K	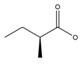	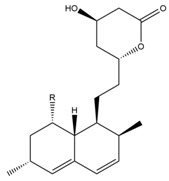	C_24_H_36_O_5_	404	Lipid lowering	[32]
J	OH	C_19_H_28_O_4_	320	Lipid lowering
L	H	C_19_H_28_O_3_	304	Lipid lowering
X	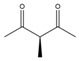	C_24_H_34_O_6_	418	Lipid lowering
M	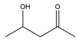	C_23_H_34_O_6_	406	Lipid lowering
K acid form	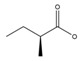	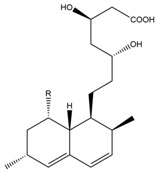	C_24_H_38_O_6_	422	Lipid lowering
J acid form	OH	C_19_H_30_O_5_	338	Lipid lowering
L acid form	H	C_19_H_30_O_4_	322	Lipid lowering
X acid form	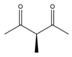	C_24_H_36_O_7_	436	Lipid lowering
M acid form	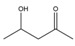	C_23_H_36_O_7_	424	Lipid lowering
T	-CH_3_	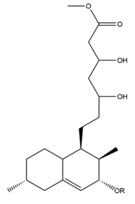	C_21_H_36_O_5_	368	Anti-tumor	[37]
U	-CH_2_CH_3_	C_22_H_38_O_5_	382	Anti-tumor
Monacolin	O	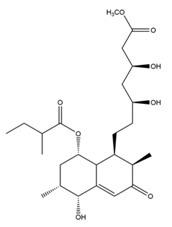	C_25_H_40_O_8_	468	Anti-tumor	[35]
P	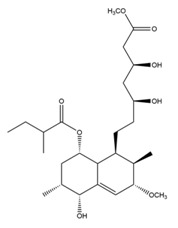	C_26_H_44_O_8_	484	Anti-tumor
Q	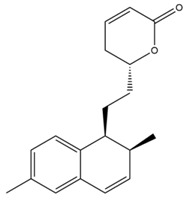	C_19_H_22_O_2_	282	Anti-tumor	[15]
S	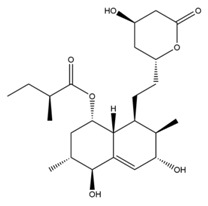	C_24_H_38_O_7_	438	Anti-tumor
R	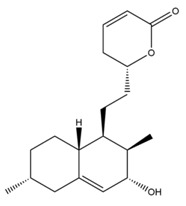	C_19_H_28_O_3_	304	Anti-tumor
Dehydro-monacolin	K	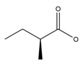	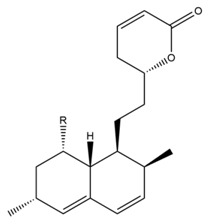	C_24_H_34_O_4_	386	Lipid-lowering Anti-tumor	[32]
N	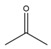	C_21_H_38_O_4_	344	Anti-tumor	[36]
L	H	C_19_H_26_O_2_	286	Anti-tumor
J	OH	C_19_H_26_O_3_	302	Anti-tumor
Dihydro-monacolin	K	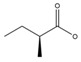	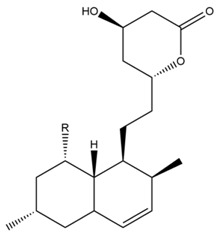	C_24_H_38_O_5_	406	Lipid-lowering Anti-tumor	[36]
L	H	C_19_H_30_O_3_	306	Lipid-lowering	[32]
α,β-dehydrodihydro-monacolin	K	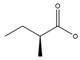	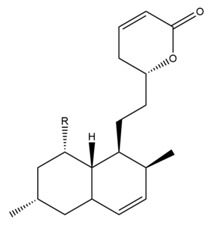	C_19_H_28_O_2_	288	Anti-tumor	[36]
L	H	C_24_H_36_O_4_	388	Anti-tumor
the ethyl ester of monacolin K	-CH_2_CH_3_	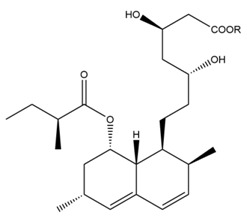	C_26_H_42_O_6_	450		[36]
the methyl ester of the hydroxyl acid form of monacolin K	-CH_3_	C_25_H_40_O_6_	436	
6*a-O-*ethyl-4,6-dihydro-monacolin L	-CH_2_CH_3_	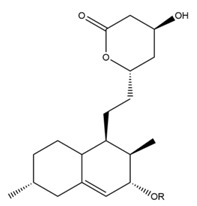	C_21_H_34_O_4_	350	Anti-tumor	[37]
6*a*-*O*-methyl-4,6-dihydro-monacolin L	-CH_3_	C_20_H_32_O_4_	336	Anti-tumor
3α-hydroxy-3,5-dihydro-monacolin L		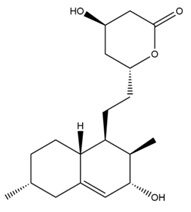	C_19_H_30_O_4_	322	Lipid-lowering	[15]
3β-hydroxy-3,5-dihydro-monacolin L	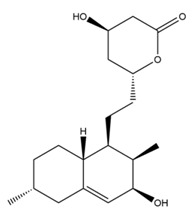	Lipid-lowering
α,β-dehydro-monacolin S		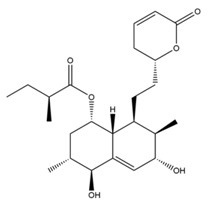	C_24_H_36_O_6_	420	inhibition of cancer cell proliferation	[15]
α,β*-*dehydro-monacolin Q		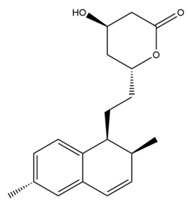	C_19_H_24_O_3_	300	inhibition of cancer cell proliferation
Compactin		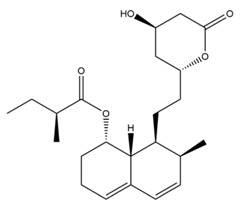	C_23_H_34_O_5_	390	Lipid-lowering	[32]
Monacophenyl		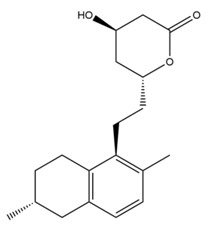	C_19_H_26_O_3_	302		[34]
Dihydro-monacolin MV		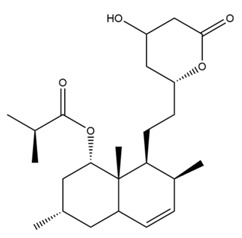	C_24_H_38_O_5_	406	2,2-diphenyl-1-picrylhydrazyl (DPPH) radical scavenging activity	[33]
(1*S*,2*S*,4*aR*,6*S*,8-*S*,8*aS*,3*′S*,5*′R*,2″S)-methyl-1,2,4a,5,6,7,8,8a-octahydro-3′,5′-dihydroxy-2,6-dimethyl-8-[(2-methyl-1-oxobutyl)oxy]-1-naphthale-neheptanoate		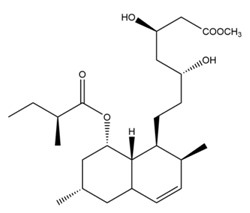	C_25_H_42_O_6_	438		[36]

**Table 2 molecules-24-01944-t002:** The detection methods of monacolin K of RYR.

Detection Methods	Detection Conditions			Detection Index	Ref.
	Mobile Phase	Column	Detector		
HPLC	Acetonitrile:Water (45:55, *v*/*v*, pH was adjusted to 2.5 with H_3_PO_4_); 1 mL/min	Nucleosil C-18	UV detector; 238 nm	Monacolin K (lactone)	[45]
HPLC (Shimadzu)	Acetonitrile:10 mM Phosphate buffer solution (60:40, *v*/*v*, pH was adjusted to 3.5 with 10% H_3_PO_4_); 1 mL/min	Millennium Sil C-18 (125 × 4.6 mm, 5 μm)	UV-visible detector; 236 nm	Monacolin K (lactone)	[46]
HPLC (Shimadzu 10A)	Methanol:0.1% Phosphate (72:28, *v*/*v*, isocratic elution for 25 min, linear elution with 100% methanol until 30 min); 0.5 mL/min; 40 °C	Wakosil-II 5C18 (250 × 4.6 mm, 5 μm)	UV detector; 238 nm	Monacolin K (lactone) and citrinin	[47]
HPLC (Agilent 1200)	Acetonitrile:Water (55:45, *v*/*v*, pH was adjusted to 2.5 with H_3_PO_4_); 1 mL/min	AZORBAX SB C-18 (250 × 4.6 mm, 5 μm)	UV detector; 238 nm	Monacolin K (lactone)	[48]
HPLC (Waters 1525)	Acetonitrile:Water (55:45, *v*/*v*, pH was adjusted to 2.5 with H_3_PO_4_); 1 mL/min	ZORBAX SB C-18 (250 × 4.6 mm, 5 μm)	UV detector; 238 nm	Monacolin K (lactone)	[49]
HPLC (Waters)	Acetonitrile:0.1% H_3_PO_4_ (70:30, *v*/*v*); 1 mL/min; 25 °C	Reverse-phase (RP)-18 (250 × 4.6 mm, 5 μm)	diode-array and UV detector; 237 nm	Monacolin K (lactone)	[50]
HPLC (Varian)	Acetonitrile:Water:Methanol (5:3:1, *v*/*v*); 1 mL/min	Varian C18 (250 × 4.6 mm, 5 μm)	UV detector; 230 nm	Monacolin K (lactone)	[28]
HPLC	Acetonitrile:Water(72:28, *v*/*v*); 0.5 mL/min	Beckman Ultrasphere ODS column (150 × 4.6 mm)	UV detector; 238 nm	Monacolin K (lactone)	[51]
HPLC (Waters 2690 Alliance)	Acetonitrile (A): 0.1% Trifluoroacetic acid (B); A was from 35 to 75% in 20 min and keeping 75% from 20 to 28 min; 1 mL/min; 30 °C	Waters Symmetry C18 (150 × 3.9 mm, 5 μm)	996 PDA detector; 237 nm	Monacolin K, J and L (their lactone and acid)	[52]
HPLC (Agilent 1260)	0.1% Trifluoroacetic acid (A):Acetonitrile (B); in 20 min from 40% B to 75% B, 10 min 75% B, from 75 to 100% B in 5 min, 3 min at 100% B, from 100 to 40% B in 2 min and 5 min 40% B; 1 mL/min; 20 °C	RP-C18 (250 × 4.6 mm, 5 μm)	DAD detector; 237 nm	Monacolin K (acid and lactone)	[53]
HPLC (Dionex ultimate 3000)	Acetonitrile:50 mM KH_2_PO_4_ pH 3.5 (60:35, *v*/*v*); 1.5 mL/min; 40 °C	Dionex octadecyl silyl silica gel column (250 × 4.6 mm, 5 μm)	PAD (Ultimate 3000) detector; 237 nm	Monacolin K (acid and lactone)	[54]
Statin Window HPLC	Methanol (A):0.1% Phosphorus acid (B); Linear gradient elution from 60 to 90% of solvent A in 32 min and kept at 90% of solvent A for 3 min; 1 mL/min	-	UV detector; 237 nm	Monacolin K and L	[55]
HPCE (ACS 2000)	30–80 mM Gly-NaOH buffer (pH 10.5) containing 16% (*v*/*v*) ethanol; 16 kV; 22 °C	fused-silica capillary (51 cm × 75 μm)	UV–vis detector; 238 nm	Monacolin K (acid)	[56]
MISPE-SIA-UV	Ammonium acetate buffer (pH 4.0):Acetonitrile (90:10, *v*/*v*)	miniaturized column (5.0 × 2.5 mm)	UV detector; 240 nm	Monacolin K (lactone)	[57]
QCM nanosensor	Trichloroacetic acid (diluted with glycine–sodium hydroxide buffer at PH 10); 1 mL/min	-	QCM chip with lovastatin imprinted polymer	Monacolin K (lactone)	[58]
Voltammetric (Metrohm)	Frequency, 200 Hz; potential step, 2 mV; amplitude, 25 mV by applying a negative-going potential scan from −0.6 to −1.6 V; 22 °C	-	Auto-lab potentiostat	Monacolin K (lactone)	[59]
LC–MS/MS (Agilent 6460)	A:4 mM of ammonium formate plus 0.05% formic acid in water B:4 mM of ammonium formate plus 0.05% formic acid in methanol Gradient elution started with 50% of B, increased to 90% of B within 5 min and held for 8 min, then changed to 50% of B within 2 min.; 0.4 mL/min	Intersil ODS-3 (150 × 2.1 mm, 3.5 μm)	ESI	Monacolin K (lactone)	[60]
LC–MS (Agilent Model 1100 and MSD Esquire 3000^+^)	Acetonitrile (A):0.1% phosphoric acid (B); 35% to 75% of A for 30 min and kept at 75% of A for 5 min; 0.5 mL/min	Hypersil gold column (150 × 4.6 mm, 5 μm)	UV detection; 237 nm	Monacolin K and Dehydromonacolin K (their lactone and acid)	[61]
GC–MS (Agilent 6890)	Ultrapure helium; 1 mL/min; an initial temperature of 70 °C for 4 min, and increased by 2 °C/min 70 to 100 °C (held 2 min), Then, the temperature was varied from 100 to 200 °C at 5 °C/min (held 20 min) and increased to 280 °C (held 5 min) at 10 °C/min.	fused silica capillary column HP-5MS (30 m × 025 mm)	5975 GC/MSD mass selective detector	Monacolin K (acid and lactone)	[62]
HPLC-MS(Waters 2695 Alliance, Thermo Finnigan)	75% Methanol; 30 °C	Merck LiChroCART RP-18 (250 × 4.6 mm, 5 μm)	2996 PDA detector; 238.6 nm	Monacolin K (lactone)	[63]
HPLC-MS (Agilent HP 1100 and MSD VL model)	Acetonitrile (A): 0.1% trifluoroacetic acid (B); Linear gradient elution from 35 to 75% of solvent A in 30 min and kept at 75% of solvent A for 5 min; 1 mL/min; 35 °C	Hypersil ODS (250 × 4 mm, 5 μm)	PDA detector; 237 nm	Monacolin K (lactone and acid) and M, etc.	[64]
HPLC-MS (Waters 2695 Alliance)	Acetonitrile (A): 0.1% Trifluoroacetic acid (B); A was from 5 to 75% in 15 min, kept at 75% for 5 min, increased to 95%, then reduced to 5% in another 10 min	Waters Symmetry C18 (150 × 3.9 mm, 5 μm) and arrow-bore reversed-phase Zorbax SB-C18 (100 × 2.1 mm, 5 μm)	ESI; 2996 PDA detector; UV spectrum; 232, 239, 248 nm	Monacolin K (lactone and acid)	[65]
HPLC-MS (Hitachi, Japan)	Acetonitrile:0.2% Formic acid (70:30, *v*/*v*); 1 mL/min	Biosil ODS column (150 × 4.6 mm, 5 μm)	ESI MRM mode; 238 nm	Monacolin K (lactone and acid)	[66]
HPLC-MS (Waters 510)	Acetonitrile:Water (77:23, *v*/*v*, pH 3.0); 0.8 mL/min; room temperature	Waters Symmetry C18 (250 × 4.6 mm, 5 μm)	UV200 detector	Monacolin K (lactone, acid and its methyl ester)	[67]
HPLC-MS/MS (Thermo Fisher Scientific)	0.1% HCOOH (A):CHCN (B); 80% A form 0 to 1.0 min, 10% A from 1 to 6.0 min, 10% A from 6 to 7.5 min, 80% A from 7.5 to 8 min; 0.25 mL/min	RP- C18 (20 × 2.1 mm, 3 μm)	ESI detector	Monacolin K (lactone and acid)	[68]
MISPE-UHPLC–MS/MS (Waters)	0.5 mM Ammonium acetate (A, PH 4.0):Acetonitrile (B); B was from 30 to 70% over 3.7 min and decreasing 30% in 4.0 min; 0.35 mL/min; 40 °C	analytical column BEH C18 (50 × 2.1 mm, 1.7 μm)	ESI detector	Monacolin K (acid and lactone)	[69]
LC/DAD/FLD/MS^n^ (Agilent Series 1100)	Acetonitrile:water:formic acid (10:90:0.1, A):Acetonitrile:water:formic acid (90:10:0.05, B); 40−70% B (0−7 min) and 70−90% B (7−10 min); 1 mL/min; 25.0 ± 0.1 °C	Zorbax SB-C18 (250 × 4.6 mm, 5 μm)	ESI, UV (237nm) fluorimetric (331, 500 nm)	Monacolin K M, L, and citrinin, etc.	[70]
UHPLC–DAD–QToF-MS	0.1% Formic acid (A): Acetonitrile with 0.1% formic acid (B); 65% A–35% A in 15 min and in next 3 min to 100% B; 0.35 mL/min; 35 °C	Agilent Zorbax SB-C18 RRHD (150 × 2.1 mm, 1.8 μm)	UV, 237 nm; ESI + ve mode	Monacolin K, J, and citrinin, etc.	[71]
HPLC-Chip-QTOF-MS (Agilent 1260 capillary C, Agilent 6520)	0.1% (*v*/*v*) Formic acid (A): Acetonitrile containing 0.1% (*v*/*v*) Formic acid (B) 0–2 min, 20% B; 2–10 min, 20–25% B; 10–22 min, 25–30% B; 22–30 min, 30–35% B; 30–36 min, 35–45% B; 36–53 min, 45–60% B; 53–60 min, 60–70% B; 60–85 min, 70–90% B; 85–90 min, 90% B; 1 mL/min; 25 °C	Zorbax 80SB-C18 bonded stationary phase (150 × 4.6 mm, 5 μm)	MS/MS detection (the collision energy 120 was set at 25 eV.)	Monacolin K (acid and lactone), O, Q, and M, etc.	[38]
UHPLC-DAD–Q/TOF-MS(Agilent)	0.1% Formic acid/water solution (1/1000, A): Formic acid/acetonitrile solution (1/1000, B); 0–3 min, 53% B; 3–5 min, 53–70% B; and 5–6 min, 70% B; 0.7 mL/min; 30 °C	Agilent Zorbax SB C18 (150 × 2.1 mm, 1.8 μm)	DAD-Q/TOF-MS detector; 237 nm	Total monacolins	[72]
UPLC-QTOF-MS/MS	-	HSS C18 column (150 × 2.1 mm, 1.8 μm)	ESI	Monacolin K (lactone and acid)	[73]
400 MHz ^1^H-NMR	370 μL of distilled water and 60 μL of pH 7.4 NMR buffer (1.5 M KH_2_PO_4_ in D_2_O, 0.1% 3-(trimethylsilyl)-propionate acid-*d*_4_ (TSP), 3 mM NaN_3_)	5-mm SEI probe an Automatic Sample Changer B-ACS 120	Ultrashield spectrometer (300.0 K.)	Total monacolins	[74]

**Table 3 molecules-24-01944-t003:** The standards of quality control for RYR.

Standard	Type	Detection Methods	Indicator Component	Regulation	Ref.
Martindale Pharmacopoeia	Hongqu	No request	No request	No request	[79]
Chinese Pharmacopoeia 2015	Chinese Herbal Medicine (Hongqu)	HPLC (Cosmosil 5C18-MS-II, 26 cm × 4.6 mm, 5 μm, Methanol:Water (75:25, *v*/*v*), UV detector 237 nm)	Monacolin K (lactone)	≥0.22%	[80]
	Chinese patent medicine (Xuezhikang)	HPLC (Cosmosil 5C18-MS- II, 26cm × 4.6 mm, 5 μm, Methanol:Water (75:25, *v*/*v*), UV detector 237 nm)	Monacolin K (lactone)	Not less than 2.5 mg per capsule	
Functional red yeast rice QB/T 2847-2007	Functional food	RP-HPLC (C18, 250 mm × 4.6 mm, Methanol:Water: Phosphoric acid (385:115:0.14, *v*/*v*), UV detector 238 nm)	Monacolin K (lactone and acid)	The sum of monacolin K lactone and acid ≥0.40%	[81]
The standard of Chinese herbal medicine of Yunnan province (2005)	Chinese Herbal Medicine (Hongqu)	HPLC (Cosmosil 5C18-MS- II, Acetonitrile-Methanol-0.1% Phosphoric acid (60:5:35, *v*/*v*), UV detector 238 nm)	Monacolin K (lactone)	≥0.40%	[82]
The standard of Chinese herbal medicine of Fujian province (2009)	Chinese Herbal Medicine (Hongqu)	No request	No request	No request	[83]
The standard of Chinese herbal medicine of Henan province (1991)	Chinese Herbal Medicine (Hongqu)	No request	No request	No request	[84]
The standard of Chinese herbal medicine of Hubei province (2009)	Chinese Herbal Medicine (Hongqu)	No request	No request	No request	[85]
The standard of Chinese herbal medicine of Beijing (1998)	Chinese Herbal Medicine (Hongqu)	No request	No request	No request	[86]
The standard of Chinese herbal medicine of Shandong Province (2012)	Chinese Medicine Yinpian (Hongqu Mi)	No request	No request	No request	[87]
Standard for Chinese Medicine Yinpian Processing of Sichuan Province (2015)	Chinese Medicine Yinpian (Hongqu)	HPLC (Cosmosil 5C18-MS- II, Acetonitrile-Methanol-0.1% Phosphoric acid (55:5:40, *v*/*v*), UV detector 238 nm)	Monacolin K (lactone)	≥0.40%	[88]
Standard for Chinese Medicine Yinpian Processing of Zhejiang Province (2015)	Chinese Medicine Yinpian (Hongqu)	HPLC (Cosmosil 5C18-MS- II, Acetonitrile-Methanol-0.1% Phosphoric acid (55:5:40, *v*/*v*), UV detector 238 nm)	Monacolin K (lactone and acid)	Total ≥ 0.30%; The peak area of acid monacolin K is not less than 5% of the lactone monacolin K peak area	[89]
Standard for Chinese Medicine Yinpian Processing of Hunan Province (2010)	Chinese Medicine Yinpian (Hongqu)	No request	No request	No request	[90]
Standard for Chinese Medicine Yinpian Processing of Heilongjiang Province (2012)	Chinese Medicine Yinpian (Hongqu)	No request	No request	No request	[91]
Standard for Chinese Medicine Yinpian Processing of Hebei Province (2003)	Chinese Medicine Yinpian (Hongqu)	No request	No request	No request	[92]
Standard for Chinese Medicine Yinpian Processing of Beijing (2008)	Chinese Medicine Yinpian (Hongqu)	No request	No request	No request	[93]
Standard for Chinese Medicine Yinpian Processing of Chongqing (2008)	Chinese Medicine Yinpian (Hongqu)	No request	No request	No request	[94]
Standard for Chinese Medicine Yinpian Processing of Shanghai Province (2008)	Chinese Medicine Yinpian (Hongqu)	No request	No request	No request	[95]
Standard for Chinese Medicine Yinpian Processing of Shandong Province (2012)	Chinese Medicine Yinpian (Hongqu Mi)	No request	No request	No request	[96]
Standard for Chinese Medicine Yinpian Processing of Henan Province (2005)	Chinese Medicine Yinpian (Hongqu Mi)	No request	No request	No request	[97]
Standard for Chinese Medicine Yinpian Processing of Tianjin (2018)	Chinese Medicine Yinpian (Hongqu Mi)	No request	No request	No request	[98]
Standard for Chinese Medicine Yinpian Processing of Fujian Province (2012)	Chinese Medicine Yinpian (Hongqu Mi)	No request	No request	No request	[99]

**Table 4 molecules-24-01944-t004:** The authentication methods of functional RYR.

Detection Methods	Detection Conditions	Detection Index	Ref.
UHPLC–QQQ-MS, UHPLC-Q-TOF-MS	ACQUITY UHPLC BEH C_18_ (100 mm × 2.1 mm, 1.7 μm); A:0.1% formic acid in water; B:0.1% formic acid in acetonitrile; 0–12 min, 20–80% B; 12–14 min, 80–100% B; 14–16 min, 100% B; 0.35 mL/min; 40 °C; Q-TOF-MS detector	heptaketide	[8]
HPLC	Spherisord ODS-2 column (250 × 0.4 mm, 0.5 μm + precolumn Zorbax Reliance Cartridge); A = 0.2% phosphoric acid in water; B = acetonitrile. Gradient: A/B 65/35 to 25/75 in 20 min; A/B 25/75 to 25/75 in 28 min; 1 mL/min; DAD detector, 237 nm	Monacolin K (lactone and acid), other monacolins	[100]
Stable isotope ratio analysis. (^13^C-NMR)	The ^13^C/^12^C ratio was measured (around 0.5 mg) using an isotope ratio mass spectrometer following total combustion in an elemental analyzer; MS detector	^13^C/^12^C ratio	[101]

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
