# Peer review of "Quality and Authenticity Control of Functional Red Yeast Rice—A Review"

_molecules, 2019, doi:10.3390/molecules24101944_

Round 1

Reviewer 1 Report

Although in principle the topic could be quite interesting, extensive editing of English language and style is required because some sentences are very difficult to understand. This impairs very much the readability of the manuscript, that in my opinion is one of the most important feature a review should have. In addition, in my opinion there is an excessive amount of “technical” data reported in table form, that are not properly reflected and discussed in the main text. Finally, the subject is presented in such a way to appear too limited in scope and confined to Chinese regulation issues only (see Table 3). For all these reasons, I can not recommend pubblication of the manuscript in the present form.

Author Response

[1] Although in principle the topic could be quite interesting, extensive editing of English language and style is required because some sentences are very difficult to understand. This impairs very much the readability of the manuscript, that in my opinion is one of the most important feature a review should have.

Answer: Thank you very much for your positive comments, and we have sent our paper to the English editing company of MDPI’s language service to polish our wording.

[2] In addition, in my opinion there is an excessive amount of “technical” data reported in table form, that are not properly reflected and discussed in the main text.

Answer: Thank you very much for your positive comments. Regarding the “technical” data reported in table, we added a discussion in Part 2 “The detection methods of monacolin K of RYR” and Part 3 “The standards of quality control for RYR”. In Part 2, for researches into red yeast rice (RYR) products, HPLC is a common detection method and most laboratories and manufacturers can satisfy this condition. And we mainly added a discussion about HPLC methods. There are many HPLC methods, but the measurement conditions such as mobile phase, separation column and detector are different for each research purpose, resulting in different detectable components. So, we added the discussion about measurement conditions of HPLC methods, such as mobile phase, column, and detectors; and for different compounds, the conditions will also be changed.

In Part 3, we added a discussion about quality standards and safety checks of RYR. Firstly, we found most standards have no clear requirement for inspection of the contents of RYR, some of the above criteria clarify the required content of monacolin K lactone in RYR (generally not less than 0.40%), while just a few standards mentioned the content requirements of monacolin K lactone and acid. And secondly, based on the safety checks of RYR, citrinin and aflatoxin should be detected. Citrinin can causes serious health problems such as liver and kidney disease, nervous system damage, and carcinogenicity. Aflatoxins are potent carcinogens that may affect all organ systems, particularly the liver and kidneys, and are also genotoxic and may cause birth defects in children. It can cause immunosuppression and may reduce resistance to infectious agents such as HIV and tuberculosis. The ingestion of citrinin and aflatoxins are harmful effects to humans and animals. Therefore, it is necessary to carry out a safety check on the RYR.

[3] Finally, the subject is presented in such a way to appear too limited in scope and confined to Chinese regulation issues only (see Table 3).

Answer: Thank you very much for your positive comments. RYR is often used as food, dietary supplement abroad. In the manuscript, we mainly discussed functional RYR, which is for medicinal purposes. According to the literature search, only the British pharmacopoeia provides for the functional RYR standard abroad, and we have added in Line 247 of Part 3. Most of the standards are the laws of various provinces and cities in China and we listed in Table 3.

Reviewer 2 Report

1.          The authors can make an informative tabulation to list the possible markers and their ways of detection, etc, from the part of "4. The authentication methods of functional RYR"

2.          The authors should mention clearly the disadvantages of adding lovastatin to the red yeast rice in the article.

3.          In terms of DNA marker for authenticity and quality control, would those types of rice and/or types of yeast, be regulated using their respective DNA detection method? Additional paragraph or related research may be mentioned in the article.

4.           “many researches showed that there are many other monacolins, including dihydromonacolin MV, monacophenyl, monacolin O and P etc. in RYR, and they have the similar effects with monacolin K by pharmacological experiments” if there is any comprehensive studies evaluating respective efficacy of other monacolins, vs monacolin K?  if yes, would there be any better one, for using as both chemical marker and biological marker?

5.          The authors should reduce the coverage of traditional medicine theory in the abstract.

6.          It is advised that an extended view of conclusion may be added.

7.          The authors should check for their use of English by professionals.

Author Response

[1] The authors can make an informative tabulation to list the possible markers and their ways of detection, etc, from the part of "4. The authentication methods of functional RYR"

Answer: Thank you very much for your positive comments. We have already added the form in Part 4 "The authentication methods of functional RYR", and listed the detection methods, conditions and indicators.

[2] The authors should mention clearly the disadvantages of adding lovastatin to the red yeast rice in the article.

Answer: Thank you very much for your positive comments. We have added the risk of adding lovastatin at the Line 165 of Part 1 “The synthetic pathway and sources of monacolin K” and the Line 136 of Part 5 ”Future Perspectives and Conclusions”.

[3] In terms of DNA marker for authenticity and quality control, would those types of rice and/or types of yeast, be regulated using their respective DNA detection method? Additional paragraph or related research may be mentioned in the article.

Answer: Thank you very much for your positive comments. DNA marker can identify Monascus, but according to a market research on the manufacturers of red yeast rice (RYR) products. Some companies will conduct high temperature sterilization for RYR products after fermentation. During the processing, the gene of Monascus will be degraded. Therefore, we cannot extract DNA of Monascus from the RYR products to be detected by DNA marker technology.

And more, monacolin K generated by fermentation of screened Monascus. However, besides of strains, fermentation time, charge, amount, initial water content, inoculation amount, content of carbon source and nitrogen source have a great influence on monacolins production, and there are studies can prove that. Su et al. used Monascus purpures CCRC 31615, found that the highest production of monacolin K was at the fermentation temperature at 30 , the lowest production of monacolin K was at the fermentation temperature at 25 ℃. (Su, Y.-C., Wang, J.-J., Lin, T.-T., Pan, T.-M. Production of the secondary metabolites γ-aminobutyric acid and monacolin K by Monascus. J. Ind. Microbiol. Biotechnol. 2003, 30, 41–46.). Zhang et al. utilized Monascus ruber, found that the production of monacolin K was increasing with the fermentation time increasing to a certain degree. Monacolin K can be detected about 0.5mg/g in the third day and the content of it increased sharply for 17mg/g from day 6 to 18. (Zhang, B.-B., Xing, H.-B., Jiang, B.-J., Chen, L., Xu, G.-R., Jiang, Y., Zhang, D.-Y. Using millet as substrate for efficient production of monacolin K by solid-state fermentation of Monascus ruber. J. Biosci. Bioeng. 2018, 125, 333–338.).

The above indicate that the DNA marker technology based on strains is not appropriate enough for quality control of RYR.

 [4] “many researches showed that there are many other monacolins, including dihydromonacolin MV, monacophenyl, monacolin O and P etc. in RYR, and they have the similar effects with monacolin K by pharmacological experiments” if there is any comprehensive studies evaluating respective efficacy of other monacolins, vs monacolin K? if yes, would there be any better one, for using as both chemical marker and biological marker?

Answer: Thank you very much for your positive comments. We added a column of activity of monacolins in Table 2. Most monacolins have a lipid-lowering effect, and also found that some monacolins has anti-tumor and anti-oxidation effects in vitro. There is no literature reported that other monacolins has a more lipid-lowering effect than monacolin K.

And in our opinion, in order to truly reflect the quality of functional RYR, according to the spectral-effect, drug supervision and administration department should establish characteristic fingerprints of functional RYR to better evaluate its quality.

[5] The authors should reduce the coverage of traditional medicine theory in the abstract.

Answer: Thank you very much for your positive comments. We deleted the sentence on the theory of traditional medicine theory in the Line 13 of Abstract.

[6] It is advised that an extended view of conclusion may be added.

Answer: Thank you very much for your positive comments. We added discussion in Part 5, including views on the lack of current standards and how to establish a better quality control standard of RYR. 1) Establishing characteristic fingerprints of functional RYR based on spectrum-effect. The spectrum-effect can establish a link between the fingerprint, and the actual efficacy, according to the results of pharmacological experiments, the chromatographic peaks related to the efficacy of the drug are found out and structurally confirmed, so as to clarify the basis of the pharmacodynamic substance. 2) Establishing the TCM quality traceability system based on the quality marker. It reflects the compatibility of TCM and modern pharmacological study, the drug effect (such as effectiveness and safety) should be demonstrated to be associated with the identified quality marker. 3) Improvement of relevant policies and regulations of functional RYR should be carried out. Relevant departments should strengthen the supervision of RYR market, strictly control the application approval of the RYR producers, stipulate the fermentation process of different types of RYR, including the production conditions such as strains, fermented grains, fermentation time, etc., and check the functional RYR on the market for quality control.

[7] The authors should check for their use of English by professionals.

Answer: Thank you very much for your positive comments, and we have sent our paper to the English editing company of MDPI’s language service to polish our wording.

Reviewer 3 Report

Quality and Authenticity Control of Functional Red Yeast Rice-A Review

In my opinion the title of the article and the content do not bring to the same point. The article deals more with the monacolin K information and not about authentication. I believe that the authors must change the title of the article for being much closer to the content.

Table 2 Are another techniques used for the RYR detection as: FTIR. There is only one technique with NMR? Are too many HPLC methods which have the same principle

Line 162 the information is well known regarding HPLC, Something new?

Being a review I recommend to the authors to update the references with more articles from 2014-2019

Author Response

[1] In my opinion the title of the article and the content do not bring to the same point. The article deals more with the monacolin K information and not about authentication. I believe that the authors must change the title of the article for being much closer to the content.

Answer: Thank you very much for your positive comments. Some researchers have found that   commercial lovastatin is artificially added to common red yeast rice (RYR) to impersonate functional RYR and achieve illegal profitability in the pharmaceutical market. The purpose of this review is to summarize the quality and authenticity control of functional RYR. In current standards, monacolin K is the main components of lipid-lowering and the quality control index of functional RYR. Therefore, in Part 1 “introduction”, we introduced physical and chemical properties, mechanism of lipid-lowering and synthetic pathway and sources of monacolin K. In Part 2 ”The detection methods of monacolin K of RYR”, we summarized the detection methods of monacolin K in RYR. To sum up, we believe that title “Quality and Authenticity Control of Functional Red Yeast Rice-A Review” is appropriate.

[2] Table 2 Are another techniques used for the RYR detection as: FTIR. There is only one technique with NMR? Are too many HPLC methods which have the same principle.

Answer: Thank you very much for your positive comments. We reviewed the relevant literature and did not find the use of FTIR techniques to determine monacolins in functional RYR. And only one NMR methods used to determine the detection of monacolin K. The methods for determining monacolins are listed on the Table 2. HPLC as a classical analytical method is used in the determination of the content of RYR products, but the measurement conditions are different according to the purpose of the experiment, and the current national standard is still based on HPLC as the main analytical means.

In Part 2, for researches into RYR products, HPLC is a common detection method and most laboratories and manufacturers can satisfy this condition. And we mainly added a discussion about HPLC methods. There are many HPLC methods, but the measurement conditions such as mobile phase, separation column and detector are different for each research purpose, resulting in different detectable components. So, we added the discussion about measurement conditions of HPLC methods, such as mobile phase, column, and detectors; and for different compounds, the conditions will also be changed.

[3] Line 162 the information is well known regarding HPLC, Something new?

Answer: Thank you very much for your positive comments. We updated the description of the HPLC features in Line 175 of Part 2.

[4] Being a review I recommend to the authors to update the references with more articles from 2014-2019.

Answer: Thank you very much for your positive comments. After re-searching literature, we added references in 2014-2019.

Reviewer 4 Report

This manuscript reviews how to evaluate the quality and authenticity control of functional red yeast rice. The purpose of this review paper seems to be valuable in functional food markets and scientific research areas. However, there are several points to be modified for a sufficient communication.

It is necessary to be verified English writing in the manuscript by native speakers. Some parts are hard to understand.

Authors concluded that content and ratio of acid and lactone monacolin K should be determined in conclusion, which should be added in abstract section. In addition, the exact content and ratio of them should be suggested by authors based on the literatures.

Monacolins indicate various compounds including monacolin K. More specific information is required when “monacolins” is written in detection index in Table 2.

In Table 1, each reference should be added to explain each type of compound, especially in monacolins. Also formats of references included in Tables 1 and 2 should be equal.

Author Response

[1] It is necessary to be verified English writing in the manuscript by native speakers. Some parts are hard to understand.

Answer: Thank you very much for your positive comments, and we have sent our paper to the English editing company of MDPI’s language service to polish our wording.

[2] Authors concluded that content and ratio of acid and lactone monacolin K should be determined in conclusion, which should be added in abstract section. In addition, the exact content and ratio of them should be suggested by authors based on the literatures.

Answer: Thank you very much for your positive comments. In Part 5 ”Future Perspectives and Conclusions”, we discussed about there are many other monacolins in RYR besides of monacolin K in Line 129, and some of them have a similar effect to monacolin K.

The current standards only use monacolin K as the sole indicator of quality control of functional red yeast rice (RYR). With the deepening of the study of functional RYR, it is found that monacolins play a synergistic role in the lipid-lowering effect of functional RYR. Therefore, we can not only use the content and ratio of monacolin K acid and lactone as the quality control index of functional RYR. In order to truly reflect the quality of functional RYR, drug supervision and administration departments should establish characteristic fingerprints of functional RYR to better evaluate its quality based on the spectrum-effect.

[3] Monacolins indicate various compounds including monacolin K. More specific information is required when “monacolins” is written in detection index in Table 2.

Answer: Thank you very much for your positive comments. We supplemented the “monacolins” of Table 2, but due to limited space, we listed three representative monacolins.

[4] In Table 1, each reference should be added to explain each type of compound, especially in monacolins. Also formats of references included in Tables 1 and 2 should be equal.

Answer: Thank you very much for your positive comments. Table 1 has caused your misunderstanding due to our negligence. We apologize for this, and we have revised Table 1.

Table 1 is the summary of the structure and activity of monacolins in RYR. Table 2 is a summary of the detection methods of monacolins in RYR, so the references in the two tables are not the same.

Round 2

Reviewer 1 Report

I appreciated authors’efforts to improve the quality of the paper, so I can now recommend pubblication of the manuscript.

Reviewer 3 Report

Accept as it is

Reviewer 4 Report

The revised manuscript seems to be modified well considering the reviewers' comments and  well describes the methods for quality and authenticity control of functional red yeast rice and its future applications.

So, in my opinition, it will be proper to be accepted.